# Suspension-Induced Stem Cell Transition: A Non-Transgenic Method to Generate Adult Stem Cells from Mouse and Human Somatic Cells

**DOI:** 10.3390/cells12202508

**Published:** 2023-10-23

**Authors:** Behzad Yeganeh, Azadeh Yeganeh, Kyle Malone, Shawn T. Beug, Robert P. Jankov

**Affiliations:** 1Molecular Biomedicine Program, Apoptosis Research Centre, Children’s Hospital of Eastern Ontario Research Institute, Ottawa, ON K1H 8L1, Canada; 2Department of Cellular and Molecular Medicine, Faculty of Medicine, University of Ottawa, Ottawa, ON K1H 8M5, Canada; 3Molecular Medicine, Peter Gilgan Centre for Research and Learning, SickKids Research Institute, Toronto, ON M5G 1X8, Canada; 4Department of Biochemistry, Microbiology and Immunology, Faculty of Medicine, University of Ottawa, Ottawa, ON K1H 8M5, Canada; 5Centre for Infection, Immunity and Inflammation, University of Ottawa, Ottawa, ON K1H 8M5, Canada; 6Department of Paediatrics, Faculty of Medicine, University of Ottawa, Ottawa, ON K1H 8M5, Canada

**Keywords:** anoikis, adult stem cells, somatic cells, reprograming, suspension-induced stem cell transition, single-cell RNA sequence

## Abstract

Adult stem cells (ASCs) can be cultured with difficulty from most tissues, often requiring chemical or transgenic modification to achieve adequate quantities. We show here that mouse primary fibroblasts, grown in suspension, change from the elongated and flattened morphology observed under standard adherent culture conditions of generating rounded cells with large nuclei and scant cytoplasm and expressing the mesenchymal stem cell (MSC) marker (Sca1; Ly6A) within 24 h. Based on this initial observation, we describe here a suspension culture method that, irrespective of the lineage used, mouse fibroblast or primary human somatic cells (fibroblasts, hepatocytes and keratinocytes), is capable of generating a high yield of cells in spheroid form which display the expression of ASC surface markers, circumventing the anoikis which often occurs at this stage. Moreover, mouse fibroblast-derived spheroids can be differentiated into adipogenic and osteogenic lineages. An analysis of single-cell RNA sequence data in mouse fibroblasts identified eight distinct cell clusters with one in particular comprising approximately 10% of the cells showing high levels of proliferative capacity expressing high levels of genes related to MSCs and self-renewal as well as the extracellular matrix (ECM). We believe the rapid, high-yield generation of proliferative, multi-potent ASC-like cells via the process we term suspension-induced stem cell transition (SIST) could have significant implications for regenerative medicine.

## 1. Introduction

Adult stem cells (ASCs), also referred to as somatic or tissue-specific stem cells, are a population of undifferentiated cells that inhabit distinct anatomical locations within specific organs or tissues [1,2]. These cells exhibit the remarkable capacity to differentiate into two or more distinct cell types, each possessing the unique characteristics and functionalities inherent to the tissue or organ of origin [1,2]. This inherent multipotency of ASCs plays a pivotal role in tissue homeostasis, regeneration, and repair, making them indispensable players in the realm of regenerative medicine and biological research. ASCs are commonly isolated and maintained under conventional adherent culture conditions, where they represent an exceptionally scarce subpopulation of non-adherent cells. These cells possess the unique ability to aggregate and form spherical structures known as "floating spheres," which serve as a hallmark of their stem cell identity. These spheres exhibit distinctive stem cell characteristics, such as self-renewal potential and the capacity to differentiate into multiple cell lineages [3,4,5,6]. Recent studies have also shown the use of suspension culture for the differentiation of murine and human epidermal stem cells for the generation of pluripotent and adult stem cell-derived organoids [7,8,9]. 

Our initial observations revealed that when fibroblasts are cultivated in suspension culture, they undergo a notable transformation in morphology and concurrently exhibit the expression of stem cell-associated markers. However, this intriguing phenomenon is transient, as these cells subsequently enter a phase of rapid apoptosis (anoikis or detachment-induced cell death). Therefore, we explored whether or not ASCs could be derived in larger numbers from suspensions of somatic cells cultured under non-adherent conditions. We initially studied mouse dermal fibroblasts (tail/ear fibroblasts, TEFs) [10] in novel suspension culture conditions together with newly formulated growth factor (GF)-enriched [11], serum-free culture media with the Rho-kinase inhibitor [12], designed to support the transformation, survival and proliferation of ASCs. 

## 2. Materials and Methods

### 2.1. Preparation of Tail/Ear Fibroblasts (TEFs)

To prepare tail/ear fibroblasts (TEFs), the tails and ears from adult mice (8–12 week-old C57BL/6, FVB and Ly6a-GFP mice on a B6;129 background) were peeled, minced into 1 cm pieces, placed on culture dishes and incubated in Dulbecco’s Modified Eagle Medium/Nutrient Mixture F-12 (Gibco™ DMEM/F-12, HEPES, Cat#11-330-032, Thermo Fisher Scientific, Waltham, MA, USA), containing 1% Penicillin and Streptomycin (Cat#15140122, Thermo Fisher Scientific, Waltham, MA, USA) and 20% heat-inactivated fetal bovine serum (FBS; Cat# F1051-500 mL, Sigma-Aldrich, Oakville, ON, Canada) for 7 days. Cells that migrated out of the graft pieces were transferred to new plates and maintained in DMEM/F-12 supplemented with 10% FBS (passage 2). TEFs from passage 3 were used for all experiments.

### 2.2. Suspension Culture Methods Causing Cell–Cell Contact

To overcome anoikis, cells were forced into physical contact using one of two methods that achieved similar results. 

(A) Hanging Drop Suspension Culture. In this method, the force of gravity encourages cell–cell contact. Cells were pretreated for one hour with the Rho-kinase inhibitor (ROCKi; Y-27632, 10 µM), then harvested via trypsinization and seeded at 100 cells per 25 µL drop of DMEM/F-12 supplemented with mouse or human EGF, mbFGF and mIGF-1 (all 20 ng/mL) for TEFs and HDFs, respectively, plus 10 µM Y-27632 on the inner side of a 100 mm tissue culture Petri dish lid. For PHHs, Williams Medium E serum-free medium containing hHGF, bFGF and hIGF-1 (all 20 ng/mL) and ROCKi (10 µM) was used, and for human keratinocytes, SFM serum-free medium supplemented with hEGF, bFGF and hIGF-1 (all 20 ng/mL) and ROCKi (10 µM) was used. The lid was carefully turned upside down (hanging drop) and placed on top of a Petri dish filled with 5 mL of sterile PBS and the cells were cultured in an inverted position in separate drops for 2 days. The hanging drops were maintained in a humidified atmosphere of 5% CO_2_ at 37 °C. Cells were monitored under a fluorescence microscope daily to detect the creation of aggregates. Images were taken using EVOS FL Auto Cell Imaging System (Thermo Fisher Scientific, Waltham, MA, USA). On day 3, aggregated cells were transferred to ultra-low-attachment (ULA) 96-well plates (Corning^®^ Costar^®^ Ultra-Low Attachment Multiple Well Plate, Cat#7007) containing 200 µL of the same medium in each well. Cells were maintained in 5% CO_2_, 21% O_2_ and 74% N_2_ at 37 °C for 7 days. Half of the culture medium was removed and replaced every other day.

***(B)*** *G-force via Centrifugation (250× g for 3 min)*. Cells were plated at 100–300 cells per well in ULA 96-well round-bottom plates (Corning^®^ Costar^®^ Ultra-Low Attachment Multiple Well Plate, Cat#7007) in 200 µL of DMEM/F-12 supplemented with mouse or human EGF, mbFGF and mIGF-1 (all 20 ng/mL) for TEFs and HDFs, respectively, plus 10 µM ROCKi on the inner side of a 100 mm tissue culture Petri dish lid. For PHHs, Williams Medium E serum-free medium containing hHGF, bFGF and hIGF-1 (all 20 ng/mL) and ROCKi (10 µM) was used, and for human keratinocytes, SFM serum-free medium supplemented with hEGF, bFGF and hIGF-1 (all 20 ng/mL) and ROCKi (10 µM) was used. Cells were maintained in 5% CO_2_, 21% O_2_ and 74% N_2_ at 37 °C for 7 days. Half of the culture medium was removed and replaced every other day.

### 2.3. Time Lapse Image Acquisition of Sphere Growth

The growth of spheres was imaged (phase-contrast and/or green fluorescence) every 60 min using an IncuCyte^®^ live cell analysis system (Essen BioScience, Ltd., Royston Hertfordshire, UK) equipped with a 20× objective over a 7-day time course.

### 2.4. Flow Cytometry

***(A)*** *For the apoptosis assay,* apoptotic cells were quantified using an Annexin V (Alexa Fluor™ 555 conjugate, Cat# A35108, Thermo Fisher Scientific, Waltham, MA, USA), in accordance with the manufacturer’s protocol. Briefly, at the desired time point, both lifted adherent cells or cells in suspension medium were collected via centrifugation at 250 RCF at 4 °C for 5 min, followed by two cold 1 × PBS washes. Cells were then resuspended in binding buffer (10 mM HEPES, pH 7.4, 140 mM NaCl and 2.5 mM CaCl₂) at a concentration of 1 × 10⁶ cells/mL. Briefly, 100 μL of the cell suspension was transferred to a 5 mL culture tube, and 5 μL of Annexin V (555) was added. Unstained cells were used as a negative control. The cells were gently vortexed and incubated in the dark for 15 min at room temperature. An additional 200 μL of binding buffer was added to the cell suspension and Cells were analyzed *using BD LSRFortessa*™ *Cell* Analyzer (BD Biosciences, Franklin Lakes, NJ, USA) to quantify percentages. 

***(B)*** *Immunophenotype Analysis Using Flow Cytometry:* Antibodies used for the immunophenotype analysis of mMSC and hMSC surface markers are listed in Appendix A. Cell subsets were stained with antibodies and the isotype control and analyzed *using BD LSRFortessa*™ *Cell* Analyzer (BD Biosciences, Franklin Lakes, NJ, USA). Collected events per sample numbered 15,000. Data were analyzed using FlowJo software. 

### 2.5. Immunofluorescence (IF) Staining

Immunofluorescence was performed as described previously [13]. Samples (cells and spheroids) were washed with PBS and fixed with 4% paraformaldehyde for 30 min at room temperature. After washing twice with PBS, cells were permeabilized and blocked in a solution of 1 × PBS containing 0.1–0.5% Triton X-100 and 1% Carbo-Free Blocking Solution (SP-5040-125, Vector Laboratories, Burlingame, CA, USA) for 1 h at room temperature. Samples were then incubated with desired primary antibodies at 4 °C overnight (the list of antibodies and concentrations used can be found in Appendix A). After washing three times with PBS, secondary antibodies (1:200, Jackson ImmunoResearch, West Grove, PA, USA) were incubated at 37 °C for 1 h at room temperature in the dark. Nuclei were stained with DAPI (Roche Life Science) for 15 min. Images were digitally captured using an epifluorescent microscope (Zeiss Axioimager M2 with Apotome 2, Carl Zeiss Microscopy GmbH, Göttingen, Germany) using appropriate filter sets. Identical images acquired with different filter sets were merged using ZEN Pro software (version 2.6, Carl Zeiss Microscopy GmbH).

### 2.6. Self-Renewal Assay

To assess whether or not cells within TEF-derived spheroids could be propagated as secondary cultures to determine their self-renewal capacity, we used an established method [3,14] with slight modifications. Spheres were transferred into a small tube and after neutralizing trypsin with media, spheres were mechanically broken down by pipetting the solution up and down with a small pipet tip. Cells were counted using a hemocytometer and trypan blue, single cells were then seeded in 96-well culture plates with 200 µL of DMEM/F-12 in each well, supplemented with GFs with a limiting dilution approach and wells containing a single cell were monitored for proliferation and clonal expansion. The frequency of expansion from single cells was calculated by dividing the number of wells containing new spheroids by the total number of wells containing a single cell [3,14]. 

### 2.7. In Vitro Differentiation Assays of Mouse Adipogenic or Osteogenic Differentiation

Fibroblasts and MSC-derived spheres were plated on a 2-well chamber slide and grown in 10% DMEM/F-12. Next day, the medium was changed to adipogenic or osteogenic differentiation media [15] (StemXVivo steogenic/Adipogenic Base Media, cat# CCM007, RD systems, Minneapolis, MN, USA) and replaced every 3–4 days for a total of 21 days. Adipocyte lipid droplets and osteoblast calcification were detected via oil red O and alizarin red S staining, respectively (Louise Pelletier Histology Core Facility, University of Ottawa)

### 2.8. Single-Cell RNA Library Preparation and Sequencing

SIST cells were prepared in accordance with the protocol described in this study. Between 70,000 and 100,000 mouse adherence-cultured fibroblasts (monolayer) and dissociated spheroid cells (single cells) were freshly prepared and their gene expression profile analyzed using single-cell 3′ RNA-sequencing (StemCore Laboratories, Ottawa General Hospital, U Ottawa). Sequencing libraries were prepared using Single Cell 3′ Reagent Kits V3.1 (10x Genomics, Pleasanton, CA, USA) with the 10x Chromium controller, and sequenced on NextSeq 500 (Illumina, San Diego, CA, USA). Library construction, sequencing and initial analysis were performed by StemCore Laboratories (Ottawa Hospital Research Institute, University of Ottawa). 

### 2.9. Bioinformatics Analysis

Sequencing data were processed with Cellranger v7.0.0 to generate cell vs. gene Unique Molecular Identifier (UMI) count matrices using the 10x Genomics mouse genome reference (refdata-gex-mm10-2020-A). Matrices were loaded into R (v4.2.1) and subsequent analysis was performed in Seurat v4.3.0. Data from the attached and suspension culture libraries were separately run through a series of quality control steps, retaining cells with at least 200 detected genes and including only genes detected in more than three cells. The data from each library were processed with scDblFinder v1.10.0 using default parameters to identify potential cell doublets, and each library was filtered to remove doublets and cells with >25% mitochondrial transcripts. UMI counts in the retained cells were normalized using the SCTransform algorithm and the libraries were integrated using the Seurat data integration pipeline with 3000 features. Principal component analysis (PCA) was run on the integrated assay, a nearest neighbor graph was constructed using the first 25 principal components and cells were clustered using the Louvain algorithm at a resolution of 0.3. Uniform manifold approximation and projection (UMAP) embedding was performed for data visualization, again using the first 25 principal components (integrated UMAP). To visualize differences between the two libraries, SCTransform was run on the raw count data from the combined libraries, followed by PCA and the conduction of a second round of UMAP embedding using the first 25 principal components of this new PCA dimensional reduction (SCTransform UMAP). Markers for each cluster were identified using the Seurat FindAllMarkers function with the Wilcox test, searching only for markers with a positive log2 fold change. To identify differentially expressed genes between pairs of clusters or between suspension and attached cells, the FindMarkers command was used.

### 2.10. Human Hepatocyte Differentiation

To assess the capacity of differentiation between PHH-derived spheroids and ductal cells, spheres were seeded in a serum-free medium, Williams Medium E, containing hHGF, bFGF and hIGF-1 (all 20 ng/mL) until they formed a monolayer. The medium was then changed to a transition and expansion medium (TEM) containing DMEM/F12 supplemented with insulin-transferrin-serine (ITS) (Cat# I3146, Sigma-Aldrich, Oakville, ON, Canada), with the following growth factors or small molecules: hEGF (20 ng/mL, Cat# 78006.1, Stemcell technologies, Vancouver, BC, Canada), hHGF (20 ng/mL, Cat# 100-39H, Peprotech, Cranbury, NJ, USA), Y27632 (10 μM, Cat# 10005583), CHIR99021 (3 μM, Cat# 13122-1), Sphingosine-1-phosphate (S1P) (1 μM, Cat#62570), lysophosphatidic acid (LPA) (5 μM, Cat#10010093-1) and A83-01 (1 μM, Cat# 9001799-5) (all from Cayman Chemical, Ann Arbor, MI, USA), for 7 days as described previously [16].

### 2.11. Tumorigenic Assessment of TEF-Derived Spheroids

Mouse fibroblast cells and spheres were suspended at 1 × 10^7^ cells/mL in DMEM/F12 containing 10% FBS. BALB/c nude mice were anesthetized and 100 μL of the cell suspension (1 × 10^6^ cells) was injected subcutaneously into the dorsal flank. Ninety days after the injection, tissues of the dorsal flank were dissected from the mice, fixed in 4% formaldehyde and embedded in paraffin. Sections underwent hematoxylin and eosin staining and were evaluated by a histopathologist (Dr. A Gutsol).

### 2.12. Statistics

All numerical data are presented as mean ± SEM from at least three separate experiments. *p*-values were obtained via a 2-tailed *t-*test for 2 groups, or a one-way analysis of variance (ANOVA) followed by a post hoc Tukey test for more than 2 groups using GraphPad Prism Version 6.0 software (GraphPad Software, San Diego, CA, USA). Differences were considered significant at *p* < 0.05.

## 3. Results

### 3.1. Expression of MSC Surface Markers by Mouse TEFs Grown in Suspension Culture While Undergoing Detachment-Induced Apoptosis (Anoikis)

Stem cell antigen-1 (Sca1; Ly6A) is a well-established marker of murine hematopoietic and mesenchymal stem cells (MSCs) [17]. We used dermal fibroblasts containing a *Ly-6A (Sca-1)* GFP (*Sca-1-*GFP) transgene to monitor the conversion of TEFs into MSCs. TEFs from 8- to 12-week-old *Sca-1-*GFP transgene mice [18] were isolated and plated (500 cells/well) on ultra-low-attachment (ULA) 96-well plates. Once detached, the elongated and flattened cells transformed into a rounded morphology with large nuclei and a scant cytoplasm (Figure 1A). Within 24h, an increase in *GFP* expression (endogenous *Sca-1*) was detected (Figure 1B–E). GFP expression was rapidly lost when cells were moved back to the adherent culture (Appendix A), suggesting the possible conversion of TEFs into MSC-like cells in the suspension. Consistent with this interpretation, flow cytometric analysis using mMSCs surface markers [15] Sca-1, CD29, CD44, CD90.1, CD105, and CD106, and CD45R as a negative marker, revealed a transition of wild-type TEFs after 24 h of suspension culture to a MSC-like phenotype (Figure 1F). However, the majority of MSC-like cells derived using this method, which we term suspension-induced stem cell transition (SIST), subsequently underwent apoptosis (anoikis) [19], as evidenced by the increased expression of Annexin V (Figure 1G,H) and cleaved Caspase 3 (Appendix A).

### 3.2. Overcoming Anoikis with Generation of Proliferative Mouse TEF-Derived Spheroids Expressing MSC Markers with Self-Renewal Capacity

It is known that pro-survival signaling pathways are activated via cell–cell and cell-matrix anchorage [20] and that detachment triggers anoikis [21]. We therefore examined whether or not a modified suspension culture, designed to favor cell–cell contact, may avoid anoikis. TEFs in suspension were brought into physical contact either via gravity, using hanging drop suspension culture (Figure 2Ai), or via centrifugation (250× *g* for 3 min) in ULA 96-well round-bottom plates (Figure 2Aii). The increase in *GFP* expression observed within 24 h in spheroids generated using both methods confirmed the activation of endogenous *Sca-1,* (Figure 2B, Appendix A). For all subsequent experiments, ULA 96-well plates were employed, as they were more suitable for live cell imaging.

To characterize wild-type TEF-derived spheroids, they were initially transferred to an 8-well chamber slide and allowed to adhere for 1 h. We first evaluate whether or not the formation of spheroids using the SIST method reduces anoikis induced via detachment by conducting IF staining using an antibody specific for cleaved caspase 3 (C.CASPASE3). The IF staining for C.CASPASE3 demonstrated the absence of apoptotic cells within the spheroids (Figure 2C). We then assessed the expression of mMSC surface markers [15] Sca-1, CD29, CD44, CD90.1, CD105, CD106 and CD45R as a negative marker via IF. As shown in Figure 2D, the majority of cells were positive for MSC markers and negative for CD45R. To evaluate proliferation capacity, spheroid volume was measured over 7 days; steady growth was observed (Figure 2E, Movie S3). To ensure our observations in TEFs were not mouse strain-dependent, spheroid formation was reproduced using TEFs from C57BL/6J and FVB/NJ wild-type murine strains (Appendix A). To further examine the proliferative capacity of spheroid cells, we stained for the proliferation marker Ki-67 (Appendix A). The proliferative index (proportion of Ki67 positive cells) was significantly increased in spheroids compared to adherent TEFs (Appendix A). Therefore, TEF-derived spheroid cells using our novel SIST platform remain viable, are highly proliferative and express MSC markers. 

We next examined the self-renewal capacity of wild-type TEF-derived spheroid cells, using a well-established assay [3]. Spheroids were enzymatically and mechanically dissociated and cells were seeded, one per well, in 96-well plates (288 wells total). The formation of clones was then evaluated after 14 days (Figure 2F). As shown in Figure 2G, a significant increase in the number of new colonies generated from spheroid cells (9.89%) compared with adherent cells (0.78%) was observed. These observations suggest that SIST spheroids comprise a heterogeneous population of MSCs and progenitor cells without self-renewal capacity. We next examined whether or not SIST-derived spheroid cells are multipotent. Individual spheres were transferred using a micropipette onto a 2-well chamber slide. After plating and sphere adherence to the slide, cells were observed to migrate away, causing the gradual loss of the three-dimensional structure over a 5-day period (Figure 3A). Double IF staining revealed that migrating cells lost their immunoreactivity for Sca-1 and CD44 (Figure 3B), similar to the loss of *GFP* expression in *Sca-1-*GFP spheroids in adherent culture (Figure 3A, bottom). After two days in culture and forming a monolayer, cells were placed either in osteogenic or adipogenic induction culture media for 21 days [15]. Consistent with the degree of differentiation, adipocyte lipid droplets and osteoblast calcification were detected using Oil Red O and Alizarin red S, respectively (Figure 3C). To assess tumorigenic potential, TEFs (control) and spheroid cells were injected into the dorsal flanks of BALB/c nude mice and monitored over three months. No tumors were observed in any mice, which was confirmed via histological examination (Figure 3D).

### 3.3. Characterizing the Global Gene Expression Profile of Mouse TEF-Derived Spheroids Compared to Monolayer Culture

To identify the transcriptional changes resulting from growth in spheroid vs. monolayer conditions, we performed single-cell RNA sequencing (scRNA-seq) on cells grown under both conditions. After performing quality control steps to remove presumed low-viability cells (>25% mitochondrial transcripts) and cell doublets, 11,515 cells in total (8022 from the spheroid culture, and 3493 from the monolayer culture) were retained for analysis. Data from the two libraries were combined using the Seurat integration pipeline to group similar cells from both libraries and the data were visualized using uniform manifold approximation and projection (UMAP). A nearest-neighbor graph was constructed using the integrated assay and clustering was performed using the Louvain algorithm at a resolution of 0.3, identifying eight clusters (Figure 4A). Coloring cells by source library in the UMAP projection shows that the integration procedure intermixed cells from the two libraries in the UMAP space, in contrast to the UMAP projection generated on the non-integrated data, in which the cells from the two libraries are separate in UMAP space (Appendix A). The cells of both libraries can also be split into populations with a high (>2500) or low (≤2500) number of detected genes, with an increased number of lower gene count cells from the spheroid culture (Appendix A).The low- and high-gene cells group separately in the UMAP space and Louvain clustering, with clusters 0, 1, 2 and 6 containing the low gene cells, and clusters 3, 4, 5 and 7 containing high-gene cells. The identification of marker genes for each cluster revealed between 233 and 1671 markers per cluster (with an adjusted *p*-value of < 0.05), with 3365 genes in total being identified as markers of at least one cluster. In these clusters, spheroid cells comprise the majority of clusters 0, 1, 2, 4 and 7 (Appendix A). The heatmap suggested differential gene expression across all clusters (Appendix A), which supports their separate distribution in UMAP (Figure 4B). As seen in Appendix A, clusters 1 and 6 group together in the top dendrogram, as do clusters 4 and 7. The analysis of differentially expressed genes (DEGs) revealed that each cluster was characterized by a specific transcriptional profile (Appendix A).

Next, we examined the expression of the MSC surface markers observed in IF staining (Figure 2C) in each cluster. Consistent with the IF results in Figure 2C, the expression levels of mouse MSC genes, *Sca-1* (*Ly6a*), *CD29* (*Itgb1*), *CD44* and *CD90.1* (*Thy1)* were high in clusters 2, 3, 4, 5 and 7 (Figure 4C and Appendix A). During the initial clustering analysis conducted at a lower resolution, cluster 4 and cluster 7 were grouped together into a single cluster. However, upon a further refinement of the analysis at a higher resolution, they emerged as distinct individual clusters. Subsequently, guided by their distinct gene expression profiles, we directed our focus towards conducting an in-depth analysis of clusters 4 and 7. Our gene expression analyses revealed that cluster 4, which includes 9% of the 8022 original spheroid cells (Figure 4D), exhibited the significantly increased expression of MSC markers compared to the rest of the cells (Appendix A, and Figure 4E). We observed that the expression of many extracellular matrix (ECM) genes, including *Col1a1*, *Col1a2*, *Col3a1*, *Fbn1*, *Lama2* and *Lama4*, was significantly greater in cells of one of more clusters among clusters 2, 3, 4, 5 and 7 (Appendix A), and that expression of some of these genes was significantly higher in spheroid cells than in attached cells. Further, their expression was significantly increased in spheroid cells of cluster 4 compared with the rest of the cells (Figure 4F). Since approximately 10% of spheroids cells showed a self-renewal property (Figure 2F), we sought to explore the genes important for stem cell self-renewal. The analysis of differentially expressed genes revealed the significantly increased expression of one or more of the genes *Notch2*, *Sox4*, *Sox9*, *Klf2* and *Foxp1* [22,23,24,25,26] in clusters 3, 4, 5 and 7 (Appendix A), but only in cluster 4 was the expression all of these genes significantly greater when compared to that of the remainder of the dataset, with that of *Sox9* and *Foxp1* also being significantly higher in spheroid vs. monolayer cells within cluster 4 (Appendix A and Figure 4G and Appendix A). The heatmap shown in Appendix A displays the expression profile of selected self-renewal and collagen genes across clusters. 

The DEGs analysis of cluster 7, which had the lowest percentage of spheroid cells among the clusters (Figure 4D), showed the significant upregulation of a family of transcription factors, *Hox* genes, including *Hoxb7*, *Hoxb9*, *Hoxb13* and *Hoxa11os* (Figure 4H and Appendix A), and *Mmp10* and *Mmp13* genes (Appendix A). *Hox* genes play a crucial role as regulators of periosteal stem cell identity, affecting fibroblast, osteoblast and progenitor cells [27]. Additionally, *Mmp* genes are implicated in extracellular matrix (ECM) remodeling, a pivotal component of the stem cell niche [28]. The heatmaps demonstrating average normalized expression per cluster of genes expressed at a significantly higher level in cluster 7 and all the *Hox* and *Mmp* genes in the dataset are presented in Appendix A. Altogether, the characterization of transcriptional profiles of the TEFs-derived spheroid cells using scRNA sequencing revealed the presence of specific clusters associated with mesenchymal stem-like cells and self-renewing capacity.

### 3.4. Generation of Spheroids with SC Properties from Cells of All Three Human Germ Layers, Mesoderm, Endoderm and Ectoderm (Human Dermal Fibroblasts, Hepatocytes and Keratinocytes, Respectively)

We next assayed SIST on primary human dermal fibroblasts (HDFs), which revealed a transition of HDFs after 24h suspension culture to a MSC-like phenotype as made evident via flow cytometric analysis using hMSCs surface markers [15]: CD73, CD105, CD106, CD146, CD166, STRO-1 and CD45 (negative marker) (Figure 5A). HDFs also showed a capacity to form a growing spheroid similar to that of mouse TEFs (Figure 5B and Appendix A). Stem cell characteristics were confirmed via IF staining using human MSC-specific surface markers [15] CD105, CD73, CD106, CD146, CD166, STRO-1 and CD19 and CD45 (negative marker), (Figure 5C). Furthermore, similar to the case of TEFs, MSC markers were lost when spheroids were transformed to an adherent monolayer (Appendix A). 

Finally, we examined whether or not other human somatic cells of endodermal (hepatocyte) or ectodermal (keratinocyte) origin could also undergo SIST. Primary human hepatocytes (PHHs) formed spheres (Figure 5D) that were less compact compared to the spheroids derived from TEF and HDF-derived MSCs and displayed a delicate and fragile nature. Consequently, a notable proportion of these spheroids partially disaggregated during the transfer process onto slides for further examination, including for immunofluorescence (IF) staining. Notably, despite some disaggregation, these spheroids still expressed human hepatic stem cell-specific surface markers CD117, CD133 and EPCAM, while testing negative for the hepatocyte marker AFP [16,29] (Figure 5E). As for mesodermal cells, stem cell markers were lost when PHH-derived spheroids formed an adherent monolayer (Figure 6A,B). To confirm that PHH-derived spheroid cells are capable of differentiation into ductal cells, spheroids were grown to form a monolayer using an established transition and expansion medium for 7 days [16] and then were showed to stain for CK19 (Figure 6C), a specific ductal cell marker [16,29,30]. Primary human keratinocytes also generated proliferating spheres (Figure 6D) which were positive for stem cell markers TfR/CD71 and p63/TP73L [31,32,33], and negative for keratinocyte markers CK1 and CK5 and CK14 [33,34] (Figure 6E). Collectively, our results confirm the transformation of mouse and primary human somatic cells (originated from all three germ layers) into ASC-like cells when subjected to unique SIST suspension culture conditions designed to avoid anoikis. 

## 4. Discussion

Herein, we report a novel suspension culture method that circumvents anoikis, thus allowing a comparatively large proportion of mouse and human somatic cells to form spheroids enriched with cells that have ASC-like properties. We believe the conditions that favor the growth of ASCs in a non-adherent culture environment resulted from increased cell–cell contact in conjunction with optimized culture media. The mechanisms via which normal cells in non-adherent culture acquire ASC characteristics remain unclear. Most primary cells are considered to be anchorage-dependent for survival with anoikis rapidly ensuing once they are detached, complicating the examination of their properties in non-adherent culture [19,21]. In contrast, adherence to a plastic substrate leads to the induction of a transcriptional and surface marker shift allowing for survival [35]. Unlike normal cells, transformed or tumorigenic epithelial cells can proliferate when non-adherent, allowing an epithelial–mesenchymal transition (EMT) with the emergence of cells with stem cell properties [36]. In contrast, the suspension culture method, herein termed SIST, encourages physical contact between cells to allow normal cells to avoid anoikis, thereby more efficiently forming spheroids containing ASC-like cells.

One of the first morphological changes we observed in cells grown in suspension culture was their transformation into clusters of round-shaped cells with large nuclei and a scant cytoplasm, possibly reflecting a mesenchymal to epithelial transition (MET). Recent studies have shown that the generation of induced pluripotent stem cells from mouse fibroblasts requires the activation of intracellular MET signals [37,38], suggesting a cooperative process between exogenous transcription factors and the extracellular micro-environment. In accordance with this, our scRNA-seq analysis revealed cellular and transcriptional modules associated with the ECM, MScs and stem cell self-renewal. Therefore, our observations raise the possibility that somatic cells are intrinsically capable of transforming into ASCs in non-adherent culture conditions by activating signaling pathways similar to those observed in MET.

Importantly, ASC-like cells generated using SIST appear to have normal a morphology while lacking tumorigenicity. Furthermore, using this strategy, the derivation of ASCs from the same tissue can effectively uphold their intrinsic cellular phenotype while substantially mitigating the risk of perturbations in their epigenetic memory. Thus, this method stands as a promising avenue for obtaining stem cells that closely mirror their tissue of origin, thereby ensuring the preservation of their characteristic epigenetic signatures. This is advantageous, given that induced cells inherit numerous components of epigenetic memory from donor tissues, which represents a potential safety concern in the clinic [39]. Such precision in epigenetic maintenance holds particular significance in the realm of regenerative medicine, as it fosters the development of cells that exhibit a heightened resemblance to their in vivo counterparts, enhancing their suitability for therapeutic applications and tissue regeneration. 

Although the results of SIST obtained from cells of mesodermal origin were reproduced in human endoderm and ectoderm cells, spheres formed by human hepatocytes and keratinocytes were less dense and smaller than those derived from mouse and human fibroblasts. Particularly noteworthy is the observation that the hepatocyte spheroids exhibited a less densely packed morphology with greater fragility than that of spheroids derived from fibroblasts. This could be due to differences in the basic growth requirements of somatic cells from diverse germ layer origins and could potentially be overcome via further modifications of the culture media. Furthermore, prior studies such as those cited [40,41] utilized 10% fetal bovine serum as a supplement for hepatocyte spheroid generation, while this study opted for a culture medium enriched with defined growth factors, including hHGF, bFGF and hIGF-1. Despite the various limitations encountered in this study, it is important to highlight that the expression of hepatocyte stem cell markers remained discernible in the spheroids that had incurred breakage. Certainly, more work is required to improve the efficiency of ASC-enriched sphere formation from somatic cells derived from diverse tissues. Nevertheless, our results suggest that normal somatic cells can represent a rapid and high-yield source of ASCs, which could have important implications for regenerative medicine.

## Figures and Tables

**Figure 1 cells-12-02508-f001:**
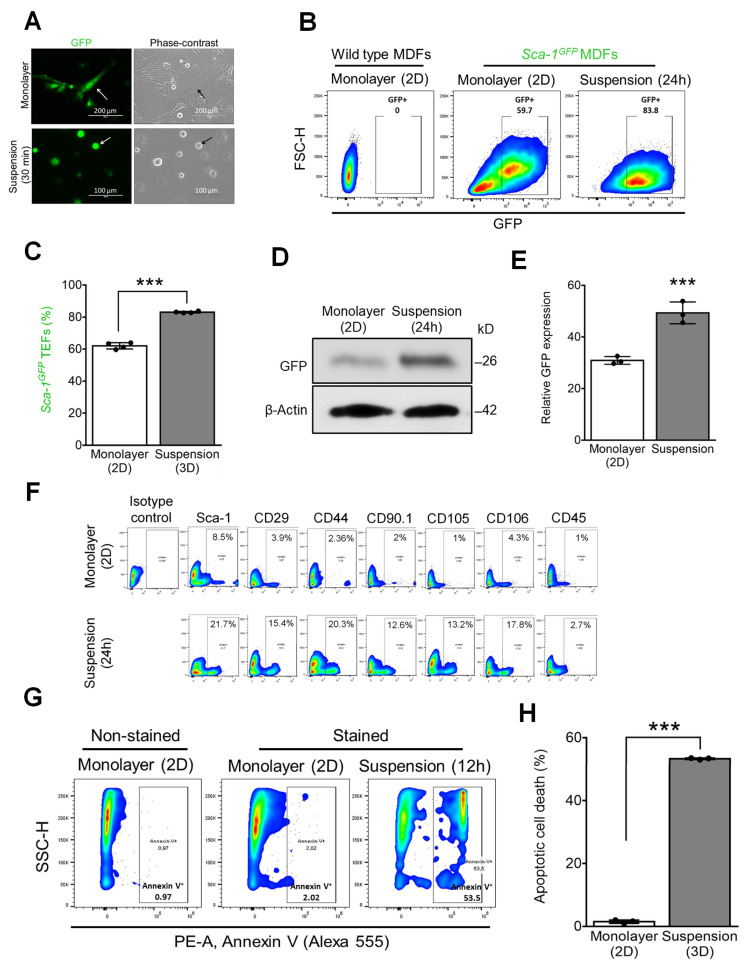
Mouse TEFs grown in suspension culture display MSC surface markers while undergoing detachment-induced apoptosis (anoikis). (**A**) Comparison of *Sca-1-*GFP TEFs morphology as a monolayer (top, pointed by white and black arrows) and after 30 min in suspension (bottom, pointed by white and black arrows). GFP fluorescence images are combined with their corresponding phase-contrast images. (**B**) Flow cytometric analysis of GFP expression of adherent wild-type or *Sca-1-*GFP TEFs cultured as a monolayer and cells cultured in suspension. (**C**) Quantitative evaluation of GFP expression of adherent *Sca-1-*GFP TEFs from (**B**). (**D**) Representative immunoblots of GFP protein in *Sca-1-*GFP TEFs grown as a monolayer and 24 h in suspension. (**E**) Densitometric analysis of GFP protein expression in monolayer *Sca-1-*GFP TEFs vs. suspension (*n* = 3). (**F**) Representative histograms for flow cytometric analysis of wild-type mouse TEFs cultured as a monolayer and 24 h in suspension. TEFs were analyzed for mMSCs surface markers, Sca-1, CD29, CD44, CD90.1, CD105, CD106 and CD45R of adherent (top) and suspension cells after 24 h (bottom). (**G**) Flow cytometry analysis for detection of Annexin V (Alexa 555) of TEFs grown for 14 h in adherent or suspension conditions. (**H**) Quantification of Annexin V-positive cells from (**F**). *** *p* < 0.001.

**Figure 2 cells-12-02508-f002:**
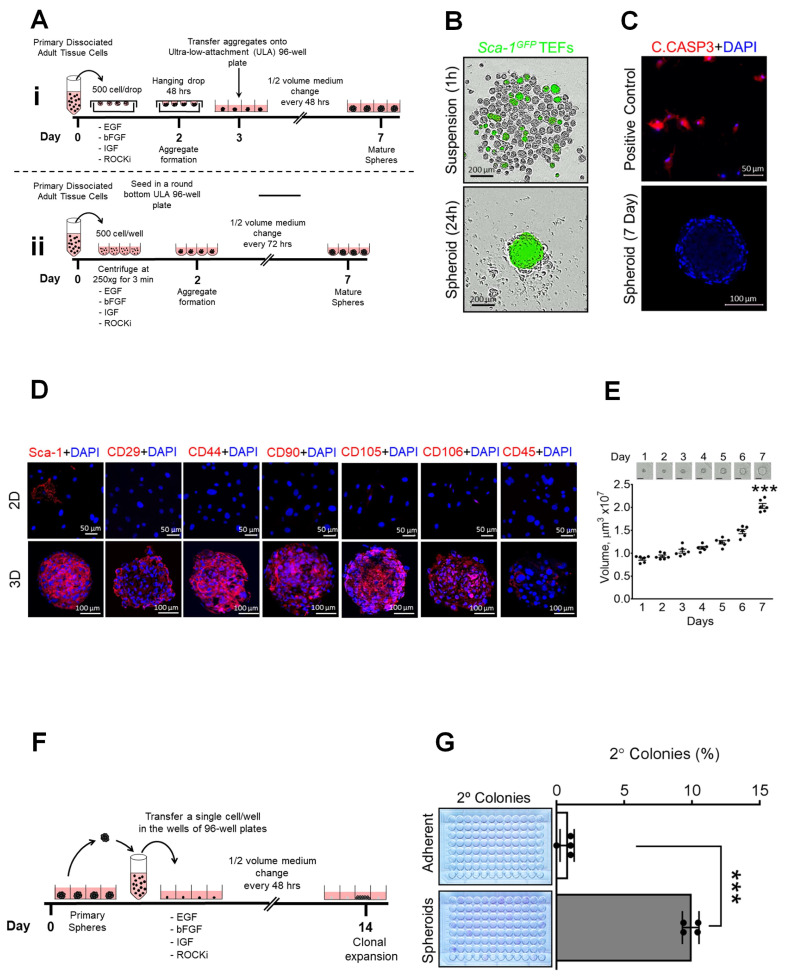
Mouse TEF-derived spheroids grown in SIST express mMSC markers, proliferate and exhibit self-renewal capacity. (**A**) Schematic representation of two suspension culture methods designed to promote cell–cell contact. (**B**) Representative microscope images captured from live cell imaging from *Sca-1-*GFP fibroblasts grown in suspension culture (left) and sphere formation (right) at 24 h. (**C**) IF staining of cleaved caspase 3 (C.CASP3) of mouse TEFs treated with 0.4 mM hydrogen peroxide as positive control (top) and mouse TEFs spheroids after 7 days (bottom). (**D**) IF staining of mouse MSCs-specific surface markers Sca-1, CD29, CD44, CD90.1, CD105, CD106 and CD45R as a negative marker of adherent cells (top) and spheroids (bottom). (**E**) Quantification of spheroid volume over a 7-day time course (*n* = 6 spheroids). (**F**) Schematic representation of methods used for self-renewal evaluation of mouse fibroblast-derived spheroids in this study. (**G**) Representative 96-well plate stained with crystal violet to identify wells with new spheroids. *** *p* < 0.001.

**Figure 3 cells-12-02508-f003:**
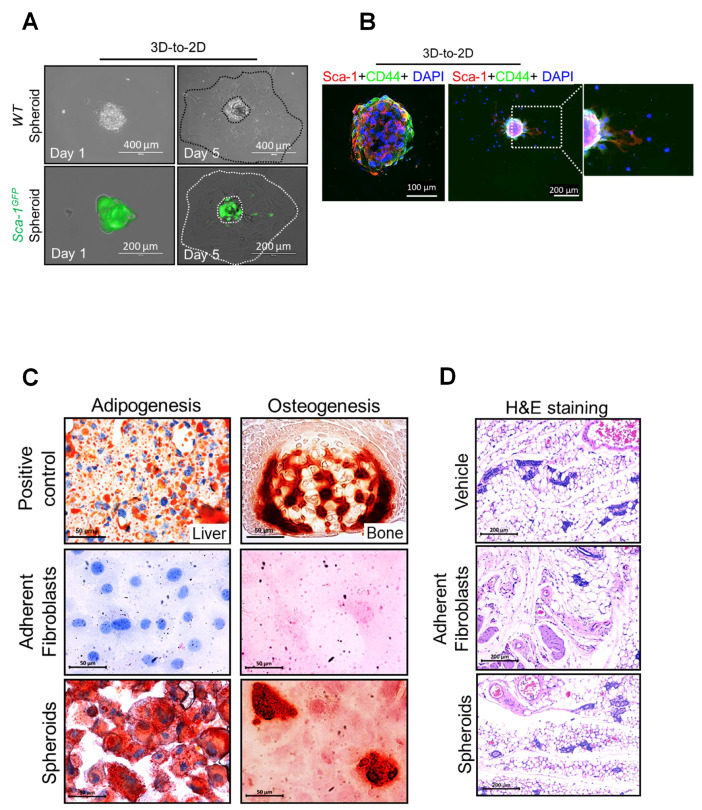
Mouse TEF-derived spheroids are multipotent with no tumorigenic potential. (**A**) Seeding of a single sphere of wild-type (top) and Sca-1-GFP (bottom) fibroblasts on a coverslip after 1 and 5 days in culture. (**B**) Co-IF staining of Sca-1 (red) with another marker of MSCs CD44 (green) in a single sphere after 1 (left panel) and 5 days (right panel) in culture. Scale bars are indicated in the images. (**C**) Representative images of Oil Red O (left) and Alizarin Red S (right) staining demonstrating adipogenic and osteogenic differentiation of adherent fibroblasts (middle panel) from TEFs derived from spheroids (bottom panel) and adherent fibroblasts (left panel). Mouse liver and bone tissue were used as positive controls (top panel). (**D**) Representative hematoxylin and eosin staining of tissues dissected from injection sites in mice receiving cell-free vehicle (Vehicle), adherent monolayer fibroblasts or spheroid-derived cells (Spheroids).

**Figure 4 cells-12-02508-f004:**
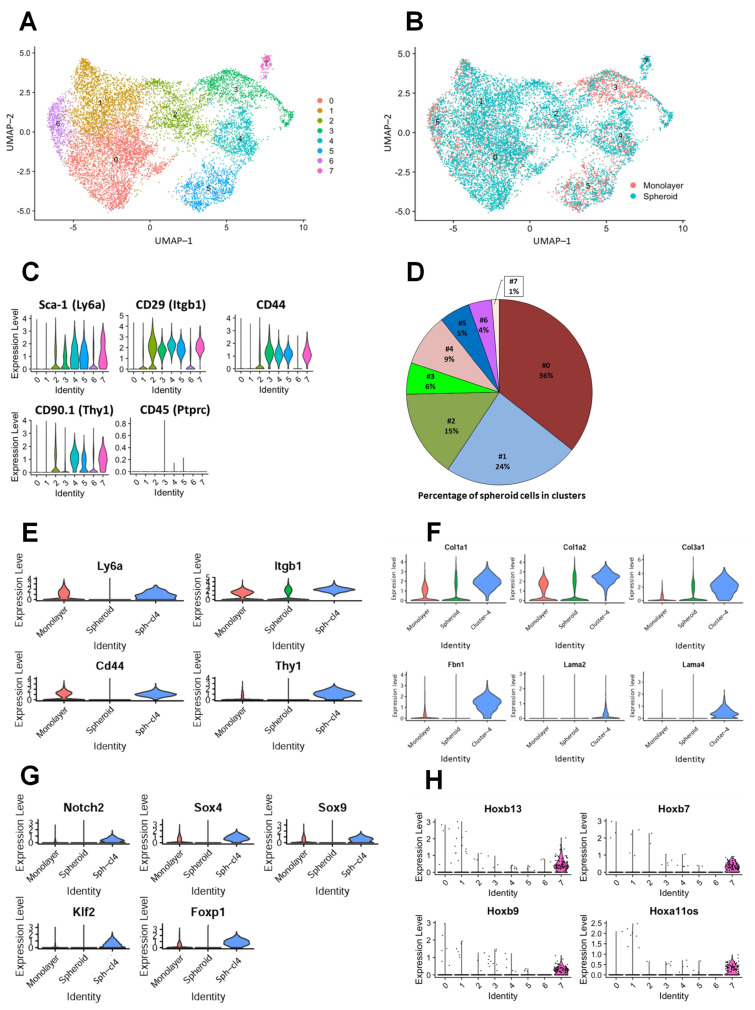
Characterization of mouse TEF-derived spheroids vs. monolayer via differential expression gene analysis. (**A**) UMAP projection of cells calculated from a principal component analysis (PCA) reduction from the Seurat integrated assay, which attempts to bring similar cells close together. Cells are colored by (**A**) cluster, identified from the integrated assay at a resolution of 0.3, or by (**B**) source library, showing the overlap of UMAP coordinates. (**C**) Violin plots of MSC surface marker genes observed in IF staining in Figure 2C, showing expression profiles split by source library within each cluster. (**D**) Pie chart demonstrating distribution of spheroid cells across clusters. (**E**) Violin plot visualization of expression of MSC surface markers genes identified as significantly enriched in cluster 4. (**F**) Violin plot visualization of expression of selected collagen, fibronectin and laminin genes identified as significantly enriched in cluster 4. (**G**) Violin plots of stem cell self-renewal genes identified as significantly enriched in cluster 4. (**H**) Violin plot visualization of expression of homeobox genes identified as significantly enriched in cluster 7.

**Figure 5 cells-12-02508-f005:**
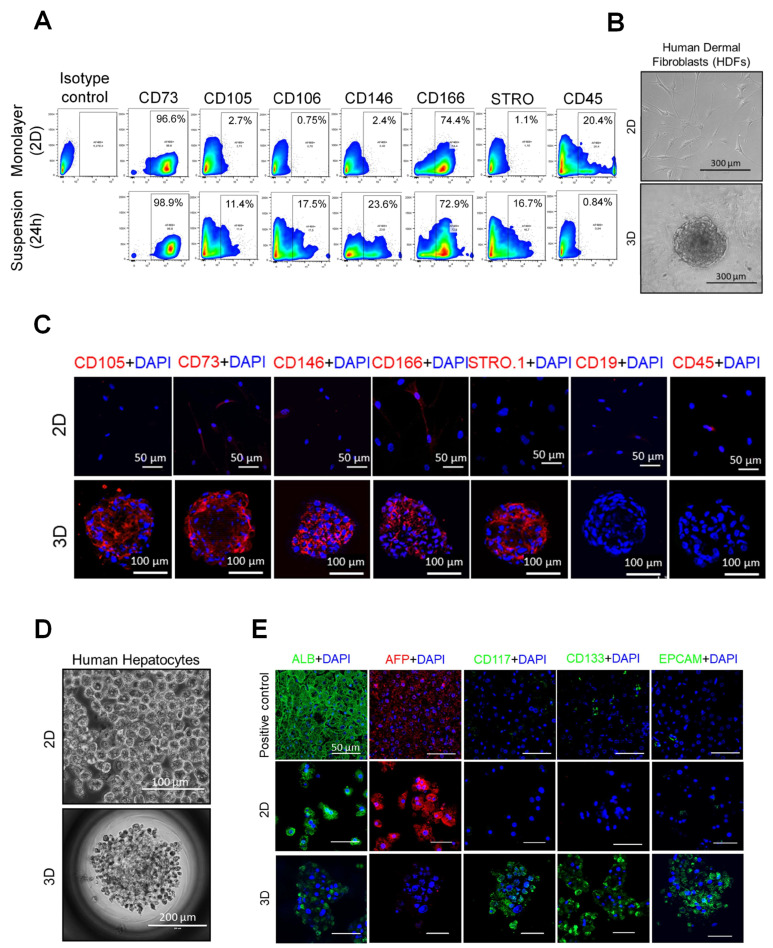
Human dermal fibroblast- and hepatocyte-derived spheroids possess SC properties. (**A**) Representative phase-contrast images of adherent monolayer human dermal fibroblasts (top) and a spheroid after 7 days in culture (bottom). (**B**) Representative histograms for flow cytometric analysis of primary HDFs cultured as a monolayer and 24 h in suspension. HDFs were analyzed for surface hMSC surface markers CD73, CD105, CD106, CD146, CD166, STRO-1 and CD45 as a negative marker of adherent (top) and suspension cells after 24 h (bottom). (**C**) IF staining of human MSC-specific surface markers CD73, CD105, CD106, CD146, CD166, STRO-1 and CD45 as a negative marker of adherent cells (top) and spheroids (bottom). (**D**) Representative phase contrast images of adherent monolayer human hepatocytes (top) and a spheroid after 7 days in culture (bottom). (**E**) Characterization of human hepatocyte-derived spheroids via IF staining using human hepatic stem cell-specific surface markers CD117, CD133, EPCAM, and AFP and ALB as a hepatocyte marker on adherent cells (top) and spheroids (bottom). Sections from human liver tissue were used as positive control (top panel). Scale bars are indicated in the images.

**Figure 6 cells-12-02508-f006:**
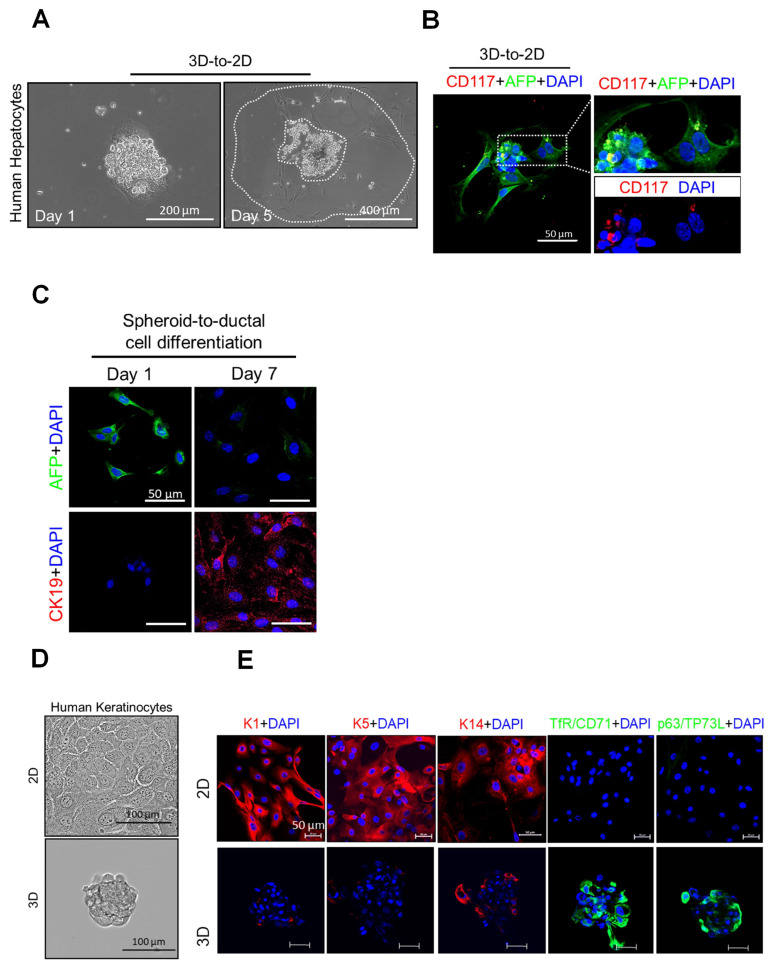
Differential capacity of human hepatocyte-derived spheroids and formation of spheroids by human keratinocytes that proliferate and express keratinocyte-specific stem cell markers. (**A**) A single sphere of human hepatocytes after seeding on coverslip at day 1 (left) and day 5 (right) in culture. (**B**) Co-IF staining of CD117 (red) with AFP (green) in a single sphere after 3 days in culture. (**C**) IF staining of AFP (green) and CK19 (red, ductal cell marker) on human hepatocytes derived from spheroids after 7 days culture in transition/expansion medium. (**D**) Representative phase-contrast images of monolayer human keratinocytes (top) and a spheroid at day 7 (bottom). (**E**) IF staining of human keratinocytes CK1, CK5 and CK14 and keratinocyte stem cell markers TfR/CD71 and p63/TP73L on adherent cells and spheroids. Scale bars are indicated in the images.

## Data Availability

Data are contained within the article. Feature barcode matrices and raw sequence data have been deposited in Gene Expression Omnibus (GEO, https://www.ncbi.nlm.nih.gov/geo/query/acc.cgi?acc=GSE241814, accessed on 29 August 2023) and are available under the accession number [czwxugschfgplub].

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
