# Peer review of "Suspension-Induced Stem Cell Transition: A Non-Transgenic Method to Generate Adult Stem Cells from Mouse and Human Somatic Cells"

_cells, 2023, doi:10.3390/cells12202508_

Round 1

Reviewer 1 Report

Yeganeh et al. investigated an innovative method that uses suspension culture to generate adult stem cells (ASCs), which derive from spheroids formed by a variety of somatic cells including mouse fibroblasts and primary human somatic cells (fibroblasts, hepatocytes and keratinocytes). Moreover, scRNA-seq was used to identify specific cell clusters with high expression of proliferative, self-renewal, MSCs-related and ECM-involved genes. Together these findings suggested a novel strategy that to generate ASCs from somatic cells with less risk of tumorigenicity.

Below are my suggestions to improve the quality of the manuscript.

1.           In the Introduction part, there are several previous studies investigating suspension culture conditions on 2-D culture and organoids using adult epithelial stem cells, cite them properly in this part. Not informative enough as an Introduction

2.           In Fig 1A, include a high magnification image that indicates “elongated and flattened

cells transformed to a rounded morphology with large nuclei and scant cytoplasm” mentioned in lines 209-210.

3.           In Fig 2B, the use of western blot or RT-qPCR to test the level of GFP and Sca-1 would be more convincing in lines 240-241.

4.           In Fig 2D, as I know that Incucyte measures the area of organoids based on 2D images, how would you generate the volume data by that?

5.           In Fig 3A, how do you define the “gradual loss of three-dimensional structure” that you mentioned in lines 274-275?

6.           In Fig 3C, the size for scale bars in positive controls is missing.

7.           In Fig 4C, consider including GO terms of MSC-related one here, same for Fig S5A, B with GO terms of ECM-related one included

8.           In Fig 4G, any references for Sox4 and Sox9?

9.           In Fig 4D, any rationale for focusing on cluster 7 suddenly? What do those upregulated genes suggest within context?

10.         In Fig 5F, would be better to include staining on the corresponding tissue section as the positive control

11.         Same as in comment 10, for Fig 6B&C, also E, it better include staining on the corresponding tissue section as the positive control

12.         In lines 445-446, “derivation of ASCs from the same tissue maintains their original cellular phenotype consistent with the maintenance of epigenetic memory”, I think the conclusion requires more experiments for further support.

Minor modifications would make the manuscript easier to read. 

Author Response

September 11, 2023

Journal of Cells,

Dear Ms. Liu

On behalf of authors of the manuscript, I would like to thank the editorials and reviewers for their insightful and constructive critiques of our research article entitled “Suspension-Induced Stem Cell Transition: A Non-Transgenic Method to Generate Adult Stem Cells from Mouse and Human Somatic Cells“.

Below is a point-by-point response to each of the reviewers’ comments. Changes are marked in red in the revised manuscript.

Sincerely,

Behzad Yeganeh, PhD

Point-by-Point Response to Reviewers’ Comments

Reviewer comments:

Reviewer #1

Yeganeh et al. investigated an innovative method that uses suspension culture to generate adult stem cells (ASCs), which derive from spheroids formed by a variety of somatic cells including mouse fibroblasts and primary human somatic cells (fibroblasts, hepatocytes and keratinocytes). Moreover, scRNA-seq was used to identify specific cell clusters with high expression of proliferative, self-renewal, MSCs-related and ECM-involved genes. Together these findings suggested a novel strategy that to generate ASCs from somatic cells with less risk of tumorigenicity.

Below are my suggestions to improve the quality of the manuscript.

  1. In the Introduction part, there are several previous studies investigating suspension culture conditions on 2-D culture and organoids using adult epithelial stem cells, cite them properly in this part. Not informative enough as an Introduction

We appreciate the comment. In the revised manuscript we included more information in introduction and provided information regarding suspension culture conditions in introduction (lines 39-58).

  1. In Fig 1A, include a high magnification image that indicates “elongated and flattened

cells transformed to a rounded morphology with large nuclei and scant cytoplasm” mentioned in lines 209-210.

We appreciate the comment. In the revised manuscript we provided high magnification images that support the statement in the text.

  1. In Fig 2B, the use of western blot or RT-qPCR to test the level of GFP and Sca-1 would be more convincing in lines 240-241.

Thanks for the valuable comment. In the revised manuscript we provided a western blot showing the level of GFP expression. Unfortunately our Sca-1 antibody did not work for Western blot.

  1. In Fig 2D, as I know that Incucyte measures the area of organoids based on 2D images, how would you generate the volume data by that?

We really appreciate the comment. The data of spheroid volumes are generated by Incocyte software. The mouse and human fibroblast generated a very uniform circular shape “spheroids”. According to the provided information, Incucutye gathers the information in 2D and calculates volume based on that assumption and the actual measured Radius of the circular image in 2D.

  1. In Fig 3A, how do you define the “gradual loss of three-dimensional structure” that you mentioned in lines 274-275?

Thanks for the valuable comment. The concept of the “gradual loss of three-dimensional structure” is grounded in the observation that, upon seeding spheroids onto a tissue culture plate, these spherical structures initially adhere to the plate and, over time, undergo a migratory process that culminates in the formation of a monolayer of adherent cells. This process represents a fundamental aspect of in vitro cell culture dynamics, where spheroid-to-monolayer transition is a well-documented phenomenon.

  1. In Fig 3C, the size for scale bars in positive controls is missing.

This comment has been addressed in the revised manuscript.

  1. In Fig 4C, consider including GO terms of MSC-related one here, same for Fig S5A, B with GO terms of ECM-related one included (Gene Ontology (GO)-terms)

Thanks for the valuable comment. GO terms of MSC-related and ECM-related are included as Supplementary Table 3 and Supplementary Table 4, respectively.

  1. In Fig 4G, any references for Sox4 and Sox9?

This comment has been addressed in the revised manuscript.

  1. In Fig 4D, any rationale for focusing on cluster 7 suddenly? What do those upregulated genes suggest within context?

Thank you very much for the valuable comment. As explained in the text, we first examined expression of the MSC surface markers observed in IF staining (Figure 2C) in each cluster and based on gene expression analyses we found that these gene expression were high in clusters 2, 3, 4, 5 and 7 however cluster 4 exhibited significantly increased expression of MSC markers compared to the rest of the cells. Clusters 4 and 7 were both grouped as a single cluster when the clustering was run at a lower resolution. Thus we decided to further explore cluster 7. The DEGs analysis of cluster 7, showed significant upregulation of a family of transcription factors, Hox genes and Mmp genes. Hox genes are shown to be important regulator of periosteal (fibroblast, osteoblast, and progenitor cells) stem cell identity (DOI: 10.1242/dev.201391). Mmp genes are also implicated in ECM (important component of stem cell niche) remodeling (DOI: 10.1002/path.1400)   

  1. In Fig 5F, would be better to include staining on the corresponding tissue section as the positive control.

Thank you very much for the valuable suggestion. In the revised manuscript we provided IF staining on the human liver sections as positive controls. Unfortunately we could not obtain human skin tissue sections to use it as positive control for Fig 6B&C, also E.

  1. Same as in comment 10, for Fig 6B&C, also E, it better include staining on the corresponding tissue section as the positive control

 Please see the answer to the previous comment.

  1. In lines 445-446, “derivation of ASCs from the same tissue maintains their original cellular phenotype consistent with the maintenance of epigenetic memory”, I think the conclusion requires more experiments for further support.

We greatly appreciate your comment. We elaborated more and made revisions to the statement in the text (lines 481-489).

Reviewer 2 Report

The authors report the development of a method to generate adult stem cells from mouse and human somatic cells.

While this approach can have many advantages, several points are however missing:

Figure 1

  • The apoptosis observed in 2D is certainly due in large part to the too low cell density and could perhaps be reduced if the cells were passaged and plated at a higher density. Indeed, the photo of the cells in 2D in fig.1A shows too few cells to be relevant whereas the fig 1C shows 60% GFP positive cells in 2D vs. 80% in suspension.
  • What about cell growth in 2D vs. Suspension?
  • The 2D cells are negative for all markers in Fig 1E. Live/dead test must be performed to ensure the cell viability. The cell surface marker analysis of cells cultured in suspension should be performed by FACS to be quantitative and not modified by the cell plating after suspension culture.
  • Proliferation of cells in 2D vs 3D in fig 1F was performed after 12 or 14h. Proliferation in 2D can have a different kinetics like a kind of quiescent phase followed by an exponential phase. The study must be performed after a longer period of time.

Figure 2

  • Fig 2B compares suspension vs spheroid cultured cells but fig 2C presents 2D cells vs spheroids. Compile the 3 conditions in the 2 analyzes.
  • Authors indicate that the formation of spheroids decreases the anoikis but no study of viability / apoptosis is presented in fig 2. Add this analysis to the figure to confirm the initial statement.
  • Mouse MSC markers studied are sca-1, CD29, CD44, and CD90.1. Add CD105 and CD106 used by the authors for hMSC to this analysis. hMSC markers studied are CD105, CD106, CD146, CD166, CD19 negative. The choice of these markers lakes CD73 and CD45 (negative) markers usually used to characterize MSCs. Complete and homogenize cell characterizations in Figures 2C and 5B.

Figures 5 and 6

  • Figure 5E: Cells in 2D do not resemble hepatocytes in culture. The other image in Fig 5E appears to show cells sedimenting more than the formation of a compact spheroid. Analysis of adhesion proteins, cutting of aggregates, or Live/dead tests must be added to ensure than the authors actually present spheroids.
  • Human hepatic stem cell markers studied are CD117, CD133 and EpCAM, which is a very surprising choice as CD117 is known as a hematopoietic stem cell marker and CD133 a marker of cancer stem cells. These choices must be better explained and the characterization must be completed with more specific markers.
  • Culture conditions used for PHH are not usual as no serum is used but S1P that has been shown to contribute to the development of liver diseases. (10.3390/ijms19030722) and LPA (to be defined) These choices must be also explained and these conditions should be validated with primary cells.
  • The legend in Fig 5F must be rectify
  • AFP is not a marker of hepatocytes but hepatoblasts. Albumin is a more specific marker of hepatocytes and its study should be added. 
  • Fig6a: Cells in right photo do not look like 2D hepatocytes. What is the viability of the cells? Are the coverstip treated or coated?
  • Fig 6C: hepatic cells that express AFP usually also express CK19. Co-labeling of both markers would be more relevant. Moreover, CK19 is also expressed by hepatoblasts and is thus not specific to ductal cells. Other markers should be added to complete the cell characterization.
  • Fig 6E: Keratinocyte stem cell markers studied are CK1, CK5, CD71 (non-specific) but no CK14 that is usually used for keratinocytes characterization. This characterization is not fully convincing.

Author Response

September 11, 2023

Journal of Cells,

Dear Ms. Liu

On behalf of authors of the manuscript, I would like to thank the editorials and reviewers for their insightful and constructive critiques of our research article entitled “Suspension-Induced Stem Cell Transition: A Non-Transgenic Method to Generate Adult Stem Cells from Mouse and Human Somatic Cells“.

Below is a point-by-point response to each of the reviewers’ comments. Changes are marked in red in the revised manuscript.

Sincerely,

Behzad Yeganeh, PhD

Point-by-Point Response to Reviewers’ Comments

Reviewer #2
The authors report the development of a method to generate adult stem cells from mouse and human somatic cells. While this approach can have many advantages, several points are however missing:

Figure 1

  • The apoptosis observed in 2D is certainly due in large part to the too low cell density and could perhaps be reduced if the cells were passaged and plated at a higher density. Indeed, the photo of the cells in 2D in fig.1A shows too few cells to be relevant whereas the fig 1C shows 60% GFP positive cells in 2D vs. 80% in suspension.

We greatly appreciate your comment and apologize for the poor-quality images, which led to confusion and errors in interpretation. We normally plate the cells at appropriate density so in the revised manuscript we provided higher quality images with same view from phase contrast and GFP channel.  

  • What about cell growth in 2D vs. Suspension?

Thank you very much for your comment. As explained in the text, cells in suspension undergo detached-induced apoptosis (Anoikis) while in 2D, they grow as adherent cells until full confluency.

  • The 2D cells are negative for all markers in Fig 1E. Live/dead test must be performed to ensure the cell viability. The cell surface marker analysis of cells cultured in suspension should be performed by FACS to be quantitative and not modified by the cell plating after suspension culture.

Thank you very much for your valuable comments. As seen in the apoptosis assay using Annexin V the rate of dead cells in 2D cells is about 2% (Fig 1E). As suggested by the respected reviewers, to quantify the MSC surface markers of cells cultured in suspension we performed flow cytometry using mouse and human MSC markers and included these data in the revised manuscript.

  • Proliferation of cells in 2D vs 3D in fig 1F was performed after 12 or 14h. Proliferation in 2D can have a different kinetics like a kind of quiescent phase followed by an exponential phase. The study must be performed after a longer period of time.

Thanks for the comment. The experiment performed in Fig 1F is apoptosis assay using Annexin V. To confirm proliferation of spheroids we measured Ki67 positive cell (Supplementary figure 2A and B).

Figure 2

  • Fig 2B compares suspension vs spheroid cultured cells but fig 2C presents 2D cells vs spheroids. Compile the 3 conditions in the 2 analyzes.

Thank you very much for the comment. Fig 2B and Fig 2C are results of two separate experiments using different cells and different messages. Fig 2B is representative microscopic images from mouse Sca-1-GFP fibroblasts in suspension culture for one hour (left) and after 24hrs of sphere formation, indicating activation of GFP or endogenous Sca-1.

Fig 2C shows characterization of spheroids which were generated using wild type mouse fibroblasts.

  • Authors indicate that the formation of spheroids decreases the anoikis but no study of viability / apoptosis is presented in fig 2. Add this analysis to the figure to confirm the initial statement.

Thank you very much for your valuable comment. To study apoptosis in spheroids, we performed IF staining using Cleaved Caspase 3 and did not detect apoptosis in the spheroids. This new data has been added in the revised manuscript.

  • Mouse MSC markers studied are sca-1, CD29, CD44, and CD90.1. Add CD105 and CD106 used by the authors for hMSC to this analysis. hMSC markers studied are CD105, CD106, CD146, CD166, CD19 negative. The choice of these markers lakes CD73 and CD45 (negative) markers usually used to characterize MSCs. Complete and homogenize cell characterizations in Figures 2C and 5B.

Thank you very much for your valuable comment. As suggested we performed IF staining and added CD105 and CD106 for mMSCs and CD73 antibodies and hMSCs characterizations. These new data have been added in the revised manuscript.

Figures 5 and 6 

  • Figure 5E: Cells in 2D do not resemble hepatocytes in culture. The other image in Fig 5E appears to show cells sedimenting more than the formation of a compact spheroid. Analysis of adhesion proteins, cutting of aggregates, or Live/dead tests must be added to ensure than the authors actually present spheroids.

Thank you very much the comment. The primary Human Hepatocytes utilized in this study were purchased from well-established companies, Zenbio and Sigma-Aldrich (Please see Table S1). We have replaced the image in Figure 5E with a better quality image. In regards to Fig 5E, as we indicated in the text “spheres formed by human hepatocytes and keratinocytes were less dense and smaller than those derived from mouse and human fibroblasts (Lines 496-498). This could be due to the fact that previous studies used 10% fetal bovine serum as supplement to the culture medium for generation hepatocyte spheroids (for example: DOI: 10.1124/dmd.120.000340, DOI: 10.1038/srep25187 and DOI: 10.1016/j.tiv.2020.105010) while in this study we used defined growth factors (hHGF, bFGF and hIGF-1). The hepatocyte spheroids in this study were not densely packed and were rather delicate and fragile. As a result, a majority of the spheroids broke when they were being transferred onto the slides for IF staining. While this study didn't encompass the detailed examination of adhesion proteins or other related experiments on hepatocyte spheroids, it's worth noting that the expression of hepatocyte stem cell markers was still noticeable in the spheroids that had broken.

  • Human hepatic stem cell markers studied are CD117, CD133 and EpCAM, which is a very surprising choice as CD117 is known as a hematopoietic stem cell marker and CD133 a marker of cancer stem cells. These choices must be better explained and the characterization must be completed with more specific markers.

Thank you very much for the comment. It is correct that CD117 and CD113 are used as markers for hematopoietic and cancer stem cells however CD117 and CD133 are also used in various tissues with different interpretation. The choice of using these markers is based on previously published studies for example in Cell Research and Gut (DOI: 10.1038/cr.2017.47 and DOI: 10.1136/gut.2005.064477) and Book chapter, “Immunohistology of the Pancreas, Biliary Tract, and Liver” by Olca Basturk, Alton B. Farris III, and N. Volkan Adsay. In book: Diagnostic Immunohistochemistry (pp.541-592). This information was used as reference for characterization of hepatocyte, hepatocyte progenitor cells as well as hepatocyte to ductal cell transition.

  • Culture conditions used for PHH are not usual as no serum is used but S1P that has been shown to contribute to the development of liver diseases. (10.3390/ijms19030722) and LPA (to be defined). These choices must be also explained and these conditions should be validated with primary cells.

Thank you very much for the comment. As explained previously, compared to previous studies where they used 10% fetal bovine serum as supplement to the culture medium, we used defined growth factors (hHGF, bFGF and hIGF-1). Sphingosine-1-phosphate (S1P, defined in the revised manuscript) is a bioactive lipid mediator which is essential for health in normal physiology, and involved in the development of certain liver diseases such as obesity-mediated liver diseases linked to obesity. Yap signaling pathway has been shown to be important in bile duct development (for a review please see: DOI: 10.1055/s-0041-1742277). Lysophosphatidic acid (LPA, defined in the revised manuscript) and S1P have been shown to activate Yap signaling (DOI: 10.1016/j.cell.2012.06.037).  LPA and S1P are also used in tissue engineering (DOI: 10.1089/ten.TEB.2015.0107) and proliferation of liver duct-like cells (DOI:10.1038/cr.2017.47).

  • The legend in Fig 5F must be rectify

This comment has been addressed in the revised manuscript.

  • AFP is not a marker of hepatocytes but hepatoblasts. Albumin is a more specific marker of hepatocytes and its study should be added. 

Thank you very much for the comment. As suggested, we added Albumin staining in the revised manuscript.

  • Fig6a: Cells in right photo do not look like 2D hepatocytes. What is the viability of the cells? Are the coverstip treated or coated?

Thank you very much the comment. As explained previously, the primary Human Hepatocytes utilized in this study were obtained from Zenbio and Sigma-Aldrich. Hepatocyte spheroid (Fig6a left) was seeded on a cover slip coated with collagen type I.

  • Fig 6C: hepatic cells that express AFP usually also express CK19. Co-labeling of both markers would be more relevant. Moreover, CK19 is also expressed by hepatoblasts and is thus not specific to ductal cells. Other markers should be added to complete the cell characterization.

Thank you very much for your valuable comment. Unfortunately we couldn’t perform Co-IF staining since both AFP and CK19 antibodies used in this study are from same species. According to the book, “Immunohistology of the Pancreas, Biliary Tract, and Liver” by Olca Basturk, Alton B. Farris III, and N. Volkan Adsay, embryonal hepatocytes contain cytokeratins (CK) 8, 18, and 19; however, mature ones contain only CK8 and 18, with CK19 being negative by the tenth week of gestation. Furthermore, intrahepatic bile ducts and peribiliary glands stain for cytokeratins 7, 8, 18, 19, 34βH11, 34βH12, and AE1/AE3. Acinar cells generally do not label with AE1/AE3 or CK7 and CK19, whereas ductal cells are strongly positive for these markers.

  • Fig 6E: Keratinocyte stem cell markers studied are CK1, CK5, CD71 (non-specific) but no CK14 that is usually used for keratinocytes characterization. This characterization is not fully convincing.

Thank you very much for your valuable comment. As suggested, we added CK14 staining in the revised manuscript. For keratinocytes characterization we used these studies (DOI: 10.1073/pnas.061032098, DOI: 10.1073/pnas.95.7.3902, DOI: 10.1073/pnas.1100332108) all published in PNAS.

Reviewer 3 Report

The presented study on the suspension culture of somatic cells, by suspension-induced stem cell transition (SIST), to generate adult stem cells (ASCs) is highly intriguing and engaging. The experimental approach of generating ASCs through SIST, coupled with the systematic assessment of their proliferative capacity, self-renewal capacity, multipotency, and transcriptional changes using single-cell RNA sequencing, is remarkable. The data presented in this study is meticulously organized and supported by a robust methodology, complemented by excellent images. I believe that this data will contribute novel insights and valuable knowledge to researchers working in the field of stem cell biology and regenerative medicine.

While the presented data in this manuscript is commendable, the authors should consider the following minor issues to improve the quality of this manuscript:

1. Figures 1E, and 5F, lack quantitative analyses, which should be included for the reader to fully evaluate the strength of the data.

2. As spheroids grow in size, a crucial limitation arises due to restricted diffusion of nutrients and oxygen within the inner regions affecting the viability and functionality of cells residing within the spheroid. How did you address this, did the authors check the viability of the spheroids after 24h or after 7 days?

3. To assess the proliferative potential of the spheroids, a time course of 7 days was conducted. Whether the spheroids maintained the expression of mesenchymal stem cell (MSC) markers throughout this period?

4. In Figure legend 5, line 387 the caption E should be changed to F.

5. Please comment on how adult stem cells generated by SIST are important for regenerative medicine.

Author Response

September 11, 2023

Journal of Cells,

Dear Ms. Liu

On behalf of authors of the manuscript, I would like to thank the editorials and reviewers for their insightful and constructive critiques of our research article entitled “Suspension-Induced Stem Cell Transition: A Non-Transgenic Method to Generate Adult Stem Cells from Mouse and Human Somatic Cells“.

Below is a point-by-point response to each of the reviewers’ comments. Changes are marked in red in the revised manuscript.

Sincerely,

Behzad Yeganeh, PhD

Point-by-Point Response to Reviewers’ Comments

Reviewer comments:

Reviewer #3
The presented study on the suspension culture of somatic cells, by suspension-induced stem cell transition (SIST), to generate adult stem cells (ASCs) is highly intriguing and engaging. The experimental approach of generating ASCs through SIST, coupled with the systematic assessment of their proliferative capacity, self-renewal capacity, multipotency, and transcriptional changes using single-cell RNA sequencing, is remarkable. The data presented in this study is meticulously organized and supported by a robust methodology, complemented by excellent images. I believe that this data will contribute novel insights and valuable knowledge to researchers working in the field of stem cell biology and regenerative medicine.

While the presented data in this manuscript is commendable, the authors should consider the following minor issues to improve the quality of this manuscript:

  1. Figures 1E, and 5F, lack quantitative analyses, which should be included for the reader to fully evaluate the strength of the data.

Thank you very much for the valuable suggestion. We performed flow cytometry using mouse and human MSC markers and included these data in the revised manuscript.

  1. As spheroids grow in size, a crucial limitation arises due to restricted diffusion of nutrients and oxygen within the inner regions affecting the viability and functionality of cells residing within the spheroid. How did you address this, did the authors check the viability of the spheroids after 24h or after 7 days?

Thank you very much for the comment. The respected reviewer is correct regarding hypoxia in spheroids. According to previous studies, tumor cells spheroids exceeding 400 μm in diameter develop a hypoxic core and activate known survival signaling pathways to maintain cell viability (DOI: 10.1186/s13046-017-0570-9). However the onset of hypoxia in mesenchymal stem cell spheroids has been shown to be at diameter of 180 μm (DOI: 10.3389/fbioe.2021.611837). In this study we maintain the size of spheroids not to exceed 180 um. For confirmation, we performed IF staining using Cleaved Caspase 3 and did not detect apoptosis in the spheroids. This new data has been added in the revised manuscript.

  1. To assess the proliferative potential of the spheroids, a time course of 7 days was conducted. Whether the spheroids maintained the expression of mesenchymal stem cell (MSC) markers throughout this period?

Thank you very much for the comment. The data of this study show that the expression of MSC markers starts when fibroblasts are in suspension (24 hrs) and during the time course of 7 days when they generate spheroids they maintain the expression of MSC markers.

  1. In Figure legend 5, line 387 the caption E should be changed to F.

This comment has been addressed in the revised manuscript.

  1. Please comment on how adult stem cells generated by SIST are important for regenerative medicine.

Thank you very much for the comment. In the revised manuscript we added few sentences that highlight the suitability of stem cells generated by SIST for therapeutic applications and tissue regeneration (lines 491-493).
